# *O*-Glycan-Altered Extracellular Vesicles: A Specific Serum Marker Elevated in Pancreatic Cancer

**DOI:** 10.3390/cancers12092469

**Published:** 2020-08-31

**Authors:** Takahiro Yokose, Yasuaki Kabe, Atsushi Matsuda, Minoru Kitago, Sachiko Matsuda, Miwa Hirai, Tomomi Nakagawa, Yohei Masugi, Takako Hishiki, Yuki Nakamura, Masahiro Shinoda, Hiroshi Yagi, Yuta Abe, Go Oshima, Shutaro Hori, Yutaka Nakano, Kazufumi Honda, Ayumi Kashiro, Chigusa Morizane, Satoshi Nara, Shojiro Kikuchi, Takahiko Shibahara, Makoto Itonaga, Masayuki Ono, Naoko Minegishi, Seizo Koshiba, Masayuki Yamamoto, Atsushi Kuno, Hiroshi Handa, Michiie Sakamoto, Makoto Suematsu, Yuko Kitagawa

**Affiliations:** 1Department of Surgery, Keio University School of Medicine, Tokyo 160-8582, Japan; t.yokose124@gmail.com (T.Y.); matsuda-sa@umin.ac.jp (S.M.); yukinakamura52@gmail.com (Y.N.); masa02114@yahoo.co.jp (M.S.); hy0624@gmail.com (H.Y.); abey3666@gmail.com (Y.A.); oshgo@hotmail.com (G.O.); shutaro.hori@gmail.com (S.H.); yutaka.nakano.9833@gmail.com (Y.N.); kitagawa@a3.keio.jp (Y.K.); 2Department of Biochemistry, Keio University School of Medicine, Tokyo 160-8582, Japan; a-matsuda@keio.jp (A.M.); mhirai@keio.jp (M.H.); t-nakagawa@z8.keio.jp (T.N.); htakako@keio.jp (T.H.); 3Department of Pathology, Keio University School of Medicine, Tokyo 160-8582, Japan; masugi@z6.keio.jp (Y.M.); msakamot@z5.keio.jp (M.S.); 4Department of Biomarkers for Early Detection of Cancer, National Cancer Center Research Institute, Tokyo 104-0045, Japan; khonda@ncc.go.jp (K.H.); akashiro@ncc.go.jp (A.K.); 5Department of Bioregulation, Graduate School of Medicine, Nippon Medical School, Tokyo 113-8602, Japan; 6Department of Hepatobiliary and Pancreatic Oncology, National Cancer Center Hospital, Tokyo 104-0045, Japan; cmorizan@ncc.go.jp; 7Department of Hepatobiliary and Pancreatic Surgery, National Cancer Center Hospital, Tokyo 104-0045, Japan; sanara@ncc.go.jp; 8Institute for Advanced Medical Science, Hyogo College of Medicine, Nishinomiya 663-8501, Japan; skikuchi@hyo-med.ac.jp; 9Department of Oral and Maxillo-Facial Surgery, Tokyo Dental College, Tokyo 102-8159, Japan; sibahara@tdc.ac.jp; 10Healthcare Business Division, JVCKENWOOD Corporation, Yokosuka 239-8520, Japan; itonaga.makoto@jvckenwood.com (M.I.); ono.masayuki@jvckenwood.com (M.O.); 11Tohoku Medical Megabank Organization, Tohoku University, 2-1 Seiryo-machi, Aoba-ku, Sendai 980-8573, Japan; nmine@med.tohoku.ac.jp (N.M.); koshiba@megabank.tohoku.ac.jp (S.K.); masiyamamoto@med.tohoku.ac.jp (M.Y.); 12The Advanced Research Center for Innovations in Next-Generation Medicine, Tohoku University, 2-1 Seiryo-machi, Aoba-ku, Sendai 980-8573, Japan; 13Graduate School of Medicine, Tohoku University, 2-1 Seiryo-machi, Aoba-ku, Sendai 980-8575, Japan; 14Biotechnology Research Institute for Drug Discovery, National Institute of Advanced Industrial Science and Technology (AIST), Tsukuba 305-8565, Japan; atsu-kuno@aist.go.jp; 15Department of Chemical Biology, Tokyo Medical University, Tokyo 160-0023, Japan; hhanda@tokyo-med.ac.jp

**Keywords:** extracellular vesicles, lectin microarray, pancreatic cancer, liquid biopsy, *O*-glycan, glycomics

## Abstract

**Simple Summary:**

Pancreatic cancer (PC) is among the most lethal malignancies due to an often delayed and difficult initial diagnosis. Therefore, the development of a novel, early stage, diagnostic PC marker in liquid biopsies is of great significance. In this study, we analyzed the differential glycomic profiling of extracellular vesicles (EVs) derived from serum using lectin microarray. The glyco-candidates of PC-specific EVs were quantified using a high-sensitive exosome-counting system, ExoCounter. An absolute quantification system for altered glycan-containing EVs elevated in PC serum was established. EVs recognized by *O*-glycan-binding lectins ABA or ACA were identified as candidate markers by lectin microarray. Quantitative analyses using ExoCounter revealed that the ABA- or ACA-positive EVs were significantly increased in the serum of PC patients. These specific EVs with O-glycans can act as potential biomarkers in a liquid biopsy for PC patients screening.

**Abstract:**

Pancreatic cancer (PC) is among the most lethal malignancies due to an often delayed and difficult initial diagnosis. Therefore, the development of a novel, early stage, diagnostic PC marker in liquid biopsies is of great significance. In this study, we analyzed the differential glycomic profiling of extracellular vesicles (EVs) derived from serum (two cohorts including 117 PC patients and 98 normal controls) using lectin microarray. The glyco-candidates of PC-specific EVs were quantified using a high-sensitive exosome-counting system, ExoCounter. An absolute quantification system for altered glycan-containing EVs elevated in PC serum was established. EVs recognized by *O*-glycan-binding lectins ABA or ACA were identified as candidate markers by lectin microarray. Quantitative analyses using ExoCounter revealed that the ABA- or ACA-positive EVs were significantly increased in the culture of PC cell lines or in the serum of PC patients including carbohydrate antigen 19-9 negative patients with high area under curve values. The elevated numbers of EVs in PC serum returned to normal levels after pancreatectomy. Histological examination confirmed that the tumors stained with ABA/ACA. These specific EVs with *O*-glycans recognized by ABA/ACA are elevated in PC sera and can act as potential biomarkers in a liquid biopsy for PC patients screening.

## 1. Introduction

Pancreatic cancer (PC) has a poor prognosis among the common malignancies [1]. PC is ranked as the fourth leading cause of cancer-related death in the United States and Japan in 2020, and the five-year survival rate of PC patients is less than 10% due to its highly aggressive nature and difficult initial diagnosis [2]. Surgical resection remains the only curative treatment for PC patients without distant metastasis, but only 20–30% of them can undergo resection at the time of diagnosis [2,3]. Diagnostic imaging technologies, such as computed tomography, magnetic resonance imaging, positron emission tomography, and endoscopic ultrasound, have improved the diagnostic accuracy and the prognosis for various tumors, but they have limited utility as screening tools for early stage PC because of the low accuracy and sensitivity (76–95% and 50–90%) [4,5,6,7,8,9]. In contrast, carbohydrate antigen 19-9 (CA19-9) is widely used as a serological marker for PC diagnosis [10,11]. However, the PC diagnosis using CA19-9 is limited by false negative results in the Lewis negative phenotype (5–10%), and false positive results in the presence of other gastrointestinal cancers and inflammatory diseases, such as common bile duct stones and chronic pancreatitis [12,13]. While several methods for detecting pancreatic cancer-derived molecules in the body fluids have recently been reported [14,15], no reliable method for early stage PC diagnosis has been established yet. Hence, the identification of a reliable screening marker is necessary to improve PC diagnosis and prognosis.

It is known that the quantitative or qualitative alteration in glycosylation is often associated with carcinogenesis and tumor progression [16,17]. CA19-9, a diagnostic marker for many gastrointestinal cancers including PC, is a representative carbohydrate antigen, sialyl-Lewis A (Neuα2-3Galβ1-3GlcNAc[α1-4Fuc]). The detection of disease-specific glycosylation changes is a potential for identifying novel biomarkers. Lectin microarray was developed as a reliable method for glycan analysis that allows global glycomic profiling in biological samples by detecting the signal pattern of the interactions between multiple lectins and carbohydrates [18,19,20]. By using the lectin microarray system, several glyco-biomarkers, such as *O*-linked glycosylation on Mucin 1 (MUC1), have been successfully identified in various diseases [21,22].

EVs are cell-secreted membranous vesicles with a diameter of 50–150 nm that contain genetic biomaterials, such as nucleic acids and proteins, and contribute to intercellular communications [23,24,25]. EVs play an important role in regulating physiological processes including metastasis of various cancer cells [26]. Several cancer-related EVs containing specific marker antigens have been reported in cancer cells or body fluids of cancer patients [27,28,29]. In PC patients, several miRNAs in EVs have been used as useful biomarkers [30]. Recently, the Glypican-1-positive EVs have been reported to be elevated in early PC serum [31]; however, other group analyses did not show any changes in glypican exosome in pancreatic cancer [32]. Furthermore, a PC-specific glycomic alteration on EVs has not been identified. Despite the importance of cancer specific EVs, their accurate quantification in body fluids, such as serum, is difficult due to the small size and containment within lipids or protein aggregates. An exosome counting system, namely ExoCounter (JVC, Yokohama, Japan), that directly detects the absolute number of the disease-specific EVs in serum, has been recently developed [33,34]. Altered glycans associated with several diseases have been reported on the surface of the EVs [35,36]. Hence, the analysis and quantification of cancer-related glycomic EVs alterations in PC sera could be a potential target for developing a novel PC-diagnostic system.

In this study, by integrating the systems of lectin microarray and ExoCounter, we have identified specific lectin-binding EVs elevated in PC sera. As schematically shown in Figure 1, we first performed differential glycomic profiling on the isolated EVs from the sera derived from 21 PC patients using lectin microarray. Then, we selected the candidate lectins that could recognize the elevated EVs in PC sera. Next, we established a novel system to quantify the *O*-glycosylated EVs elevated in PC sera obtained from two independent cohorts using the candidate lectins with ExoCounter.

## 2. Results

### 2.1. Differential Glycomic Profiling of Specific EVs in PC Sera Using Lectin Microarray

For analysis using lectin microarray, EVs in clinical serum samples were isolated using magnetic beads conjugated with a phosphatidylserine-binding protein TIM4 (MagCapture, Fujifilm, Tokyo, Japan) and were fluorescent-labeled with Cy3-succimidyl ester (Figure 1A). We confirmed that the size distribution of the isolated EVs was in a mode of 148.5 (±49.0) nm and a mean of 200.8 (±34.7) nm, and the particle count was 3.8 (±1.7) 10^8^/mL by analyses with NanoSight (Salisbury, UK) (Appendix A). The isolated EVs with MagCapture were validated using a scanning transmission electron microscope (STEM) (Appendix A). To determine the altered glycosylation on the EVs, we compared the glycoforms on the isolated EVs derived from the sera of the 21 PC patients and 10 normal controls (NC). It consisted of 45 lectin analyses (study 1). We discriminated the profiling data between the PC and the NC sera by multivariable hierarchical clustering (HCA) and principal component analyses (PCA) (Figure 2A, and Appendix A). When the mean normalized signals were compared, the glycoforms on the EVs of PC and NC partly overlapped (Figure 2B). However, the HCA showed that the exosomal glycomic profiles of the 45 lectin classified PC patients and NC into several clusters. Subsequently, the univariate analysis revealed that several lectins with a mean normalized signal >0.2 (net intensity>1000) showed a significant difference between the PC patients and NC (Table 1 and Appendix A). The signal intensities for six lectins, *Datura stramonium agglutinin* (DSA), *Solanum tuberosum lectin* (STL), *Lycopersicon esculentum lectin* (LEL), *Urtica dioica agglutinin* (UDA), *Amaranthus caudatus agglutinin* (ACA), and *Agaricus bisporus agglutinin* (ABA) were significantly elevated in the PC EVs compared to those in the NC (*p* < 0.10). The ratio between the signal intensities of tumor versus normal (T/N) was calculated as 1.375, 1.440, 1.276, 1.1170, 1.199, and 1.239 for DSA, STL, LEL, UDA, ACA, and ABA, respectively. Furthermore, we also found four lectins, *Sambucus sieboldiana agglutinin* (SSA), *Tulipa gesneriana lectin* (TXLC-I), *Calystegia sepium lectin* (Calsepa), and *phytohemagglutinin from Phaseolus vulgaris* (PHA-E), which showed lower signals on PC EVs. Moreover, the T/N ratios were calculated as 0.820, 0.764, 0.800, and 0.814 for SSA, TXLC-I, Calsepa, and PHA-E, respectively. No significant increase in the PC EVs was observed when using *Pholiota squarrosa lectin* (PhosL) or recombinant *Burkholderia cenocepacia lectin* (rBC2LCN), which contradicted the previously reported literature [37,38].

Next, we compared the lectin signals between the pre- and the post-pancreatectomy sera of five PC patients (study 2). The postoperative sera were collected before discharge at a median of 9 (8–30) days after pancreatectomy. Among the aforementioned six candidate lectins evaluated, the elevated signal intensity on ABA and ACA decreased in the EVs from the post-pancreatectomy sera, and the signals on DSA, STL, LEL, and DSA increased (Table 1). These results suggested that the ABA- or ACA-positive EVs are possibly released from the PC cells into the blood. ABA and ACA are the *O*-glycan binding lectins (e.g., core 1 structure, Galα1-3GalNAc). Several studies have shown that hyper or altered O-glycosylation is known to be associated with carcinogenesis and tumor progression [16,17,22]. In this study, we selected ABA and ACA as the probe lectins for detecting PC-derived EVs.

### 2.2. The ABA- and ACA-Positive Glycans are Strongly Localized in the Pathological Lesion of PC

To evaluate the unique glycomic feature found by lectin array analysis, histochemical staining using biotinylated ABA or ACA was performed on tissue samples obtained from the PC patients. The tissue sections of PC with the adjacent normal pancreas were used. Representative photomicrographs are shown in Figure 3. Significant staining by ABA and ACA was observed in the tumor but not in the normal tissue. The ACA-stained glycans were also observed in the precancerous lesions, such as pancreatic intraepithelial neoplasia (PanIN) and acinar-to-ductal metaplasia. ABA-stained glycans showed partial staining in the PanIN legions. These results confirmed that the ABA- and ACA-reactive glycans were strongly expressed in PC.

### 2.3. Quantification of the ABA- and ACA-Positive EVs Derived from Cancer Cell Lines with ExoCounter

We had developed the exosome-counting system, ExoCounter [33]. In this system, EVs are captured on antibody-coated optical discs and then labeled with antibody-conjugated magnetic nanobeads (FG beads). The absolute number of EVs can be measured by detecting the FG bead-EV complexes on the optical disc using an optical disc drive (Figure 1B). By using this system, we attempted to detect the ABA- or ACA-positive EVs in several human cancer cells, such as pancreatic cancer (BxPC-3 and Capan-1), colorectal cancer HCT116, breast cancer MCF7, and lung cancer A549 cells (Figure 4). First, we performed measurements using anti-CD-63 antibody (Ab) coated discs and anti-CD9 Ab conjugated FG beads with the ExoCounter. The CD63/CD9 double-positive EVs were detected at concentrations of 2,000,000–20,000,000 particles/µL (Figure 4A). Next, we attempted to analyze them using ABA- or ACA-coated discs and anti-CD9 Ab conjugated FG beads (Figure 1B). We determined that a coated lectin at a concentration of 5 µg/mL on the disc is optimal for EV detection in the BxPC-3 cell culture medium (Appendix A). Furthermore, to confirm whether the lectins specifically bind to glycans on the EVs, we performed an inhibition assay by addition of excess carbohydrates for inhibiting the binding with ABA or ACA (Appendix A). The count of the ABA- or ACA-positive EVs in BxPC-3 cell culture medium was remarkably lowered by the addition of carbohydrate inhibitors, indicating that the ABA or ACA coating on the discs specifically recognized glycans on the surface of EVs. Under similar conditions, we counted the number of EVs in the cell culture medium using ABA or ACA (disc) and anti-CD9 Ab (beads). The ABA- and ACA-positive EVs were detected in higher numbers in the culture medium of BxPC3 and Capan-1 cells at concentrations of 150,000–400,000 particles/µL compared to that of MCF7, HCT116, or A549 cells (Figure 4B). The ratio of the lectin-binding EVs from PC cells that was normalized with the number of the CD63/CD9 double-positive EVs was relatively higher among the examined cells (Figure 4C). This indicated that the ABA- and ACA-positive EVs were secreted from the PC cells.

### 2.4. The Numbers of ABA- and ACA-Positive EVs were Elevated in Sera from PC Patients

We performed the quantification of ABA- and ACA-positive EVs from 68 preoperative and 22 postoperative PC patients, and 77 NC serum samples in cohort-1. The ABA- and ACA-positive EVs were significantly higher in the preoperative PC patients than in the NC (*p* < 0.001, *p* < 0.001, respectively) (Figure 5A). Furthermore, the number of labeled EVs was significantly reduced in the post pancreatectomy sera, almost to the same level as that of NC (*p* < 0.001, *p* < 0.001, respectively) (Figure 5A). We further evaluated the PC-specific EVs using another human serum cohort (cohort-2). In this cohort, the ABA- and ACA-positive EVs were elevated in PC sera compared to that of healthy donors (*p* = 0.010, *p* = 0.064, respectively) (Figure 5B). Moreover, differential analysis of pancreatic cancer stage revealed that the ABA- and ACA-positive EVs were significantly elevated at an early stage (stage I), and the elevations showed no significant correlation with the tumor stages (Figure 5C).

### 2.5. Evaluation of the Diagnostic Utility of ABA- and ACA-Positive EV Counts Elevated in PC Sera

To evaluate the performance of ABA- or ACA-positive EV counts quantified by the ExoCounter as a diagnostic marker for PC, we performed the receptor operating characteristics (ROC) curve analysis using sera from cohorts 1 and 2. The area under curve (AUC) value using ABA or ACA was 0.838 and 0.810, respectively (Figure 6A,B). A cut-off value of 140,000 particles/µl for ABA-positive EVs showed the best separation with 77.8% sensitivity and 73.5% specificity. Likewise, a cut-off value of 77,000 particles/µl for ACA-positive EVs showed 70.1% sensitivity and 77.6% specificity (Figure 6B). Scatter plots comparing the particle counts in sera revealed a high correlation between ABA- and ACA-positive EVs (R^2^ = 0.648). This correlation was a way of separating PC patients from NC even at the early stages (Figure 6C). We also analyzed the correlation between the count of the EVs and CA19-9 levels in PC sera. The ABA- and ACA-positive EVs could be detected even in Lewis antigen-negative individuals where CA19-9 was not detectable. No correlation with CA19-9 was observed (Figure 6D). These results suggested that quantitative analysis of ABA- and ACA-positive EVs with ExoCounter was useful for PC diagnosis at an early stage, independent of the CA19-9 levels.

## 3. Discussion

The specific EVs released from cancer cells into the blood stream are considered to be an attractive target for identifying a novel diagnostic marker. Several studies have demonstrated the detection of disease-specific EVs using antibodies with various detection systems [26,28,29]. However, these measurements did not yield satisfactory data for clinically viable tests due to their quantitative and sensitivity limitations. Therefore, the accurate quantitative measurement of tumor specific EVs in blood is required for a reliable diagnosis from a liquid biopsy [28,39]. In this study, we clearly demonstrated the quantification of the elevated EVs in PC sera using specific lectins with ExoCounter. Quantification of EVs using ExoCounter presents several advantages (e.g., the ability to quantify the absolute number of EVs containing surface antigen high sensitivity and high linearity) compared to conventional methods, such as ELISA or flowcytometry [33,34]. Furthermore, ExoCounter can directly count EVs in multiple samples, such as serum without any enrichment or isolation procedure. Because of these properties, ExoCounter enabled us to quantify the specific elevation of the ABA- and ACA-positive EVs in PC sera even in early stage patients with good AUC values of the ROC curve (0.838 and 0.810, respectively). We also showed that ABA- and ACA-positive EVs were detectable even in Lewis antigen-negative sera, and their count values did not correlate with the CA19-9 values (Figure 6D). This suggests that the combined detection of these lectin-positive EVs and CA19-9 in blood could improve the diagnostic accuracy for PC in the future.

Among various glycan analytical methods, the lectin microarray is a simple and sensitive method for analyzing the glycan profile by detecting lectin binding patterns [18,19,20], leading to the discovery of glyco-biomarkers in various diseases [21,22]. While several reports have performed comprehensive glycomic analysis on the surface of the EVs with lectin microarray [40,41], these were analyzed using EVs derived from culture cells. To our knowledge, this study is the first report of a differential glycomic analysis of cancer-specific EVs derived from human sera in clinical settings.

We identified six candidate lectins, including ABA and ACA, that recognized the specific EVs elevated in PC patients compared to NC. Among them, the signals obtained using ABA and ACA, that bind to *O*-glycans (e.g., core 1 structure, Galα1-3GalNAc), decreased drastically in postoperative cases with high reliability. As the blood contains EVs derived from the tumor lesion and healthy tissues, the background value is relatively high, and, therefore, precise differential analysis is required. As the lectin array that screened the specific glycans of EVs was performed by qualitatively analysis, the relative signal difference level was low. By using the ExoCounter that can detect absolute quantification of specific EVs, we could elucidate a significant difference in PC sera for the candidate lectin (ABA and ACA)-positive EVs. In general, hyper or altered *O*-glycosylation is known to be associated with multiple steps in carcinogenesis and tumor progression [16,17,22]. In PC cells, it has been reported that the altered *O*-glycosylation on MUC1 was observed during tumor development and progression, including premalignant lesions under inflammatory conditions, but that study did not analyze it using ABA or ACA [42]. In our histochemical analyses, the ABA- or ACA-stained glycans were observed in PC, PanIN, and acinar-to-ductal metaplasia lesions, but not in the normal tissue (Figure 3). A recent report showed that glycoprotein MUC1-associated EVs increased in lung cancer sera [36]. The increase of ABA- and ACA-positive EVs might indicate the increase of MUC1-associated EVs. Although further analysis is required for elucidating detailed *O*-glycan structures and their carrier glycoproteins on the PC-specific EVs, this study suggests that the elevated ABA- and ACA-positive EVs in PC sera are attributable to the pathological alterations of the glycome in PC lesions. Besides ABA and ACA, four other lectins (DSA, STL, LEL, and DSA) were found as candidates to recognize elevated exosomes in PC serum by lectin array analysis, but these elevations did not decrease in the postoperative serum. While further studies are needed to elucidate the exact reason for this, the altered glycosylations recognized by these lectins may be associated with inflammation in PC lesions. The correlation analysis between the *O*-glycan alterations and nucleic acid information including DNAs or miRNAs is an interesting subject, and further investigation is needed in the future.

However, there are some limitations in the current study. (1) All of the samples examined are of PC with a diagnosis, and cases of undiagnosed early PC have not been verified. Further verification is required for these cases. (2) We have not examined the false-positive cases, in which CA19-9 is elevated in other diseases. Further investigation is needed using CA19-9 false positives cases with other diseases, such as chronic pancreatitis, bile duct stones, or other gastrointestinal malignancies, that display increased CA19-9 levels. (3) Although we used two independent cohorts, our participants were only Japanese, which may have resulted in potential selection bias. In addition, there was no clinicopathologic background matching between the two cohorts. (4) Serum specimens in this study were not collected from age-matched controls and PC patients.

This study demonstrated a novel quantitative detection system for glycan associated EVs in PC sera that enabled us to detect PC-specific EVs at an early stage, independent of the CA19-9 levels. While further analysis is needed, including comparison with other cancer types, our Exosome-counting system used specific lectins and has a potential to develop into a feasible diagnostic test for PC.

## 4. Materials and Methods

### 4.1. Antibody and Lectin

The following antibodies were used: anti-CD9 Ab (clone 12A12, Cosmo Bio Co. Tokyo, Japan) and anti-CD63 Ab (clone 8A12, Cosmo Bio Co., Tokyo, Japan). The following lectins were used: ABA (J-Chemical, Inc., Tokyo, Japan) and ACA (EY laboratories, CA, USA). The following biotin-conjugated lectins were used: biotinylated ABA (J-Chemical, Inc., Tokyo, Japan) and ACA (EY laboratories, CA, USA).

### 4.2. Cell Culture

The PC cell lines, BxPC-3 and Capan-1, were maintained in the Roswell Park Memorial Institute (RPMI) 1640 containing 10% fetal bovine serum (FBS) and IMEM containing 20% FBS, at 37 °C in a 5% CO_2_ humidified incubator. The colorectal cancer HCT116, breast cancer MCF7, and lung cancer A549 cells were maintained in DMEM containing 10% FBS at 37 °C in a 5% CO_2_ humidified incubator.

For analysis of EVs, semi-confluent cells (1 × 10^7^ cells/100 mm dish) were washed with phosphate buffer solution (PBS) twice and cultured in RPMI 1640 without FBS for BxPC-3, IMEM without FBS for Capan-1, and DMEM without FBS for HCT116, MCF7, and A549 cells. After incubation for 48 h, the culture media were collected and centrifuged at 13,000× *g* for 30 min. The supernatants were analyzed with ExoCounter.

### 4.3. Clinical Samples

We reviewed PC patients at Keio University Hospital from January 2014 to December 2018. The patients with received neoadjuvant treatment were excluded. Sera from 68 surgical, six inoperable PC, and 27 NC cases were obtained from Keio University Hospital. Preoperative sera were collected before surgery or chemotherapy, postoperative sera were obtained at a median of 24 (range: 8–51) days after surgery from patients who had undergone curative pancreatectomy. The blood samples (12 mL) were collected in Venoject II tubes (Terumo, Tokyo, Japan) and were centrifuged at 3000 rpm at room temperature for 10 min. Then, the sera were stored at −80 °C.

For lectin microarray analyses, preoperative sera from 15 surgical, six inoperable cases of PC, and 10 healthy donors, and postoperative sera from five surgical cases of PC were used. For quantitative analyses with ExoCounter, we used the sera from two independent cohorts of 117 PC and 98 NC.

Cohort 1: sera from surgical cases of PC (68 preoperative and 22 postoperative samples) and 27 NC obtained from Keio University hospital, and 50 NC obtained from the Tohoku Medical MegaBank Organization in Japan [43].

Cohort 2: serum samples from 49 PC patients and 21 NC obtained from prospective cohorts in the National Cancer Center Research Institute and National Center Biobank Network.

The pathological stages of cohort 1 and 2 were determined according to the eighth edition of the Union for International Cancer Control TNM (tumor, node, metastasis) classification. Clinicopathological variables, such as CA19-9, were obtained from the medical records. The sex, mean age, and pathological features of these cases are summarized in Appendix A.

All patients provided written informed consent. This study was approved by the Human Experimentation Committee of our institution and was conducted in accordance with the Helsinki Declaration of 1975. This study was also approved by the Human Experimentation Committee of Keio University Hospital (No. 20120466, No. 20130398, No. 20170086), Tohoku Medical Megabank Organization at Tohoku University (grant 2016–48), and National Center Biobank Network at National Cancer Center Research Institute (grant 2014-246, 2016-304).

### 4.4. Analysis with Lectin Microarray

The EVs were isolated from the sera of PC patients or NC in cohort-1 using the Magcapture Exosome Isolation Kit (Fujifilm) according to the manufacturer’s protocol. In brief, 0.6 mg of streptavidin magnetic beads, bound with 1 µg of biotinylated mouse Tim4-Fc, was added to the serum supplement and rotated overnight at 4 °C. The beads were washed three times with 1 mL of washing buffer (20 mM Tris-HCl, pH 7.4, 150 mM NaCl, 0.0005% Tween20, 2 mM CaCl_2_), and the bound EVs were eluted with elution buffer (20 mM Tris-HCl, pH 7.4, 150 mM NaCl, 2 mM EDTA) [44]. The size distribution of the isolated EVs was analyzed with NanoSight LM10 (Malvern Panalytical, Worcestershire, UK) as follows. The EVs were diluted 50-fold and injected in the sample chamber. All measurements were performed at room temperature for 60 s. The software used for capturing and analyzing the data was the NTA 3.2 Dev Build 3.2.16 (Malvern Panalytical). The isolated EVs were validated with STEM (S-5200, Hitachi High-Technologies Co., Tokyo, Japan) as follows. The EVs in PBS were mixed with equal amount of 4% paraformaldehyde and dropped on a Parafilm. The drop on the Parafilm was touched with a TEM grids with a collodion film for 30 min, followed by staining with 3% phosphotungstic acid for 30 s. The TEM grids were washed by deionized water and observed with a field emission scanning electron microscope. Images were acquired in scanning transmission electron mode with an acceleration voltage of 30 kV.

For fluorescence labeling of the isolated EVs, the protein concentration of EVs was measured by PierceTM BCA protein assay kit (Thermo Fisher Scientific, Inc., Waltham, MA, USA), and then the EV was adjusted to 100 ng of each protein and dissolved in 20 µL of PBS. Each 20 µL aliquot of the EVs was incubated with 10 µg of Cy3-succimidyl ester (Cy3-SE; Amersham Biosciences, Tokyo, Japan) for 60 min at room temperature in the dark. The reaction product was adjusted to 100 µL of probing buffer (Tris-buffered saline (TBS)) containing 1.0% Triton X-100 and 500 mM glycine and 1 mM CaCl_2_, 1 mM MnCl_2_), and incubated for 2 h at room temperature in the dark to inactivate the residual fluorescent reagent completely. Then, 30 µL of the Cy3-labeled EVs was further diluted twice with the probing buffer and applied on the lectin array glass slide (made-to-order LecChip™, which included PhoSL and rBC2LCN lectin spots instead of GSL-I-A4 and GSL-I-B4 lectin spots, (GP Bioscience, Co., Yokohama, Japan), and incubated overnight at 20 °C in a humid chamber. After being washed three times with probing buffer, the glass slide was scanned with an evanescent field fluorescence scanner (GlycoStation™, GP Bioscience, Co.). All data were analyzed using an Array-Pro analyzer version 4.5 (Media Cybernetics, Inc., Bethesda, MD, USA). The net intensity at each spot was calculated by subtracting the background value from signal intensities of three spots. Quantitative signals (Net Intensity <50,000) were obtained at an appropriate gain condition of the scanner. The mean fluorescence signals on triplicate spots of each 45 lectins immobilized on the array were used to normalize every lectin signal as previously reported [21,22]. Characteristics of 45 lectins on the customized LecChip are shown in Appendix A.

### 4.5. Lectin-Staining of PC Tissue Sections

The formalin-fixed paraffin-embedded (FFPE) tissue sections from 14 surgical cases of PC patients in cohort 1 were used for lectin staining. Especially, 5-μm sections of tumor or non-tumor tissue were cut from each block and placed on glass slides. Each serial section was stained with hematoxylin and eosin. After deparaffinization, the tissue sections were treated with Proteinase K (Agilent, CA, USA) and incubated for 10 min at 37 °C to enhance the lectin reactivity. After washing with deionized water, the tissue sections were incubated for 15 min in 0.5% periodic acid to deplete endogenous peroxidase. After being washed three times with deionized water, the tissue sections were incubated with blocking solution (5% bovine serum albumin (BSA)in 10 mM HEPES, pH 7.9) for ABA or 4% blockace (KAC Co., Ltd., Kyoto, Japan) in 10 mM HEPES for ACA for 60 min at room temperature in a humid chamber. Then, the sections were incubated with 5 µg/mL of biotinylated lectin dissolved in blocking solution for overnight at 4 °C in a humid chamber. After washing three times with PBS, a streptavidin–peroxidase reagent (VECTASTAIN Elite ABC Standard Kit, Vector laboratories Inc., Burlingame, CA, USA) was applied on the tissue sections and incubated for 20 min at room temperature. The tissue sections were visualized by incubating with 3,30-diaminobenzidine tetrahydrochloride reagent and counterstained with hematoxylin.

### 4.6. Preparation of Antibody-Conjugated Nanobeads

The carboxylated affinity magnetic nanobeads (FG beads) were purchased from Tamagawa Seiki (Nagano, Japan) [45,46]. The FG beads were incubated with 200 mM 1-ethyl-3-(3-dimethylaminopropyl) carbodiimide hydrochloride and N-hydroxysuccinimide in PBS (pH 7.4) for 4 h at room temperature. The beads were washed with 50 mM acetate buffer (pH 5.2) and were incubated with 1.0 g/L anti-CD9 or CD63 Ab in acetate buffer overnight at 4 °C. The beads were, then, incubated with 1 M ethanolamine in PBS for 5 h at 4 °C. The Ab-conjugated beads were washed and stored in HEPES buffer (10 mM HEPES, pH 7.9), 50 mM KCl, 1 mM EDTA, and 0.1% Tween 20 at 4 °C.

### 4.7. EV Quantification with ExoCounter

The optical disc was attached to a removal plate containing 16 wells for injecting samples. Each well was coated with 5 µg/mL lectins (ABA or ACA) or antibody (anti-CD63 Ab) in carbonate buffer (pH 9.6) for 30 min at 37 °C. After washing with PBS containing 0.05% Tween 20 (PBS-T), the disc was incubated with the blocking solution (1% BSA in PBS-T) for 30 min at 37 °C. The sample solution (12.5 µL samples diluted to 50 µL) was incubated for 2 h at 37 °C followed by washing with PBS-T. Then, approximately 1.6 × 10^8^ FG beads in blocking solution (1 µg/50 µL) were incubated for 2 min under a magnetic field. Each well was washed with PBS-T, followed by deionized water. The disc was dried with air spray for measurements with ExoCounter [33].

To determine the optimal concentration of the lectins for coating on the disc, EVs in BxPC3 cell culture medium were quantified using the disc coated with ABA at concentrations of 0–10 µg/mL. To confirm the lectin-binding with the glycans, EVs in BxPC3 cell culture medium were quantified with and without addition of 0.2 mM sialyl lactose or 0.2 M lactose for inhibiting the glycan-binding to ABA or ACA, respectively.

All serum samples were stored at −80 °C, thawed at room temperature, and centrifuged at 10,000× g at room temperature for 30 min before EV analysis. Serum samples were visually checked for hemolysis, and only hemolysis-free samples were used. For EV detection, serum samples (12.5 µL) were incubated on the discs.

### 4.8. Statistical Analysis

Statistical analysis was conducted using IBM SPSS statistics version 25.0 (IBM Corp., Armonk, NY, USA). Categorical variables were compared using the chi-square or Fisher’s exact test. Continuous variables were compared using the Student’s *T*-test, false discover rate using the Benjamini–Hochberg procedure, and the Mann–Whitney U-test with Bonferroni’s correction for multiple comparisons. We performed HCA and PCA for multivariable data using JMP Pro 11 (SAS Institute, Cary, NC, USA). In the study of lectin microarray, statistical tests were two-tailed, and values of *p* < 0.10 were considered statistically significant. All other statistical tests were two-tailed, and the level of significance was set at *p* < 0.05.

## 5. Conclusions

The EVs with *O*-glycans recognized by ABA and ACA were elevated specifically in PC sera, including those at the early stages. The significant elevation of ABA- and ACA-positive EVs in PC sera was observed independent of the CA19-9 levels at diagnosis. To our knowledge, this study is the first report of differential glycomic profiling and quantification method for cancer-specific EVs derived from human PC sera. This quantitative detection system using specific lectins has the potential to develop into a feasible diagnostic test for PC.

## Figures and Tables

**Figure 1 cancers-12-02469-f001:**
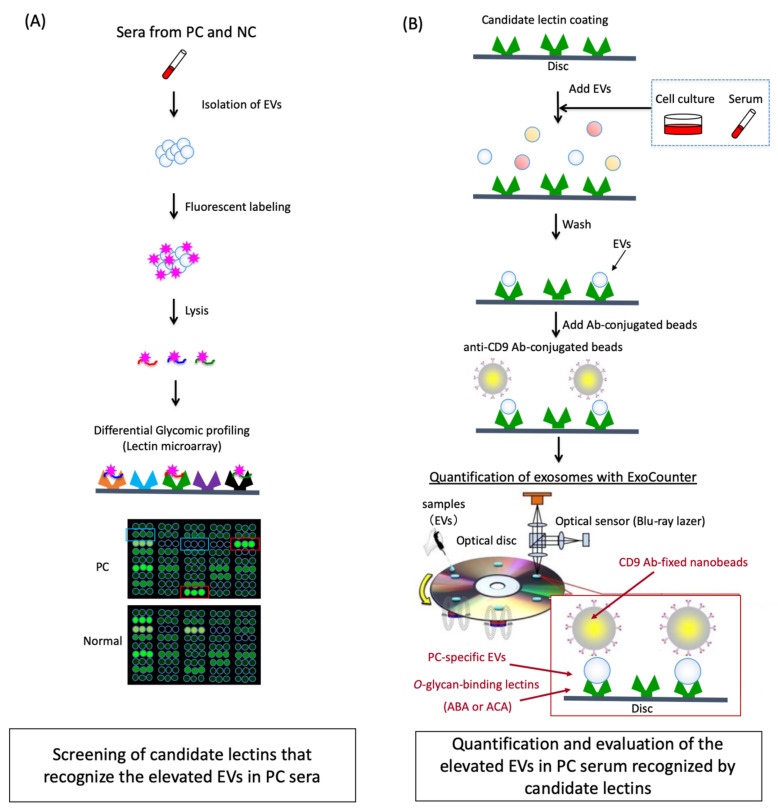
Schematic strategy for screening and quantification of elevated extracellular vesicles (EVs) with specific glycans in pancreatic cancer (PC) sera. (**A**) Screening of candidate lectins that recognize the elevated EVs in PC sera. EVs in PC or normal control (NC) sera were isolated and labeled with fluorescent tags. The glycan profiles of the labeled EVs were then analyzed using lectin microarray, and the candidate lectins that recognize the elevated EVs in PC sera were identified by comparison with NC sera. (**B**) Quantification and evaluation of the elevated EVs in PC serum recognized by candidate lectins. The candidate lectins were coated on the optical disc of the ExoCounter system. The lectin-binding EVs in the PC sera or the cell lines were captured on the disc and labeled with the anti-CD9 Ab-conjugated nanobeads. The absolute numbers of labeled-EVs were quantified using the optical disc drive of the ExoCounter. NC, normal controls; PC, pancreatic cancer; EV, extracellular vesicles.

**Figure 2 cancers-12-02469-f002:**
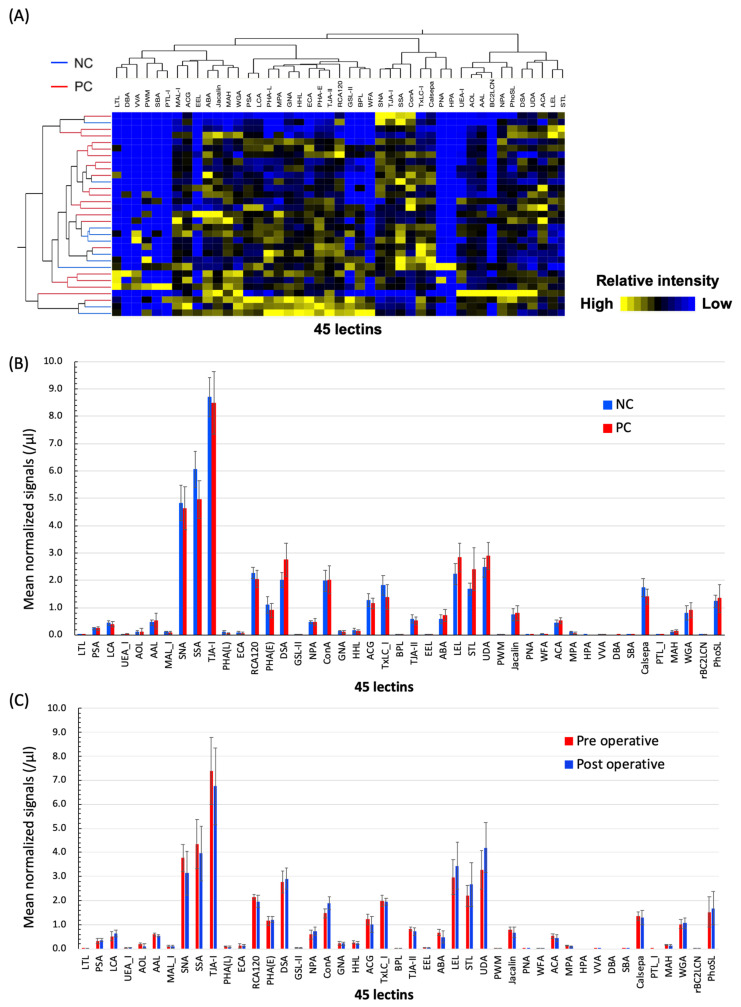
Glycomic profiling of the EVs in PC sera by lectin microarray analysis. EVs in PC (*n* = 21) or NC (*n* = 10) sera were isolated with the Magcapture Exosome Isolation Kit and then labeled with Cy3-succimidyl ester. The labeled EVs were analyzed with lectin microarray immobilized with 45 lectins. (**A**) Hierarchical clustering analysis was performed using the glycomic profiling data of PC compared to that of NC. The heat map shows a two-way cluster analysis carried out on the data of the serum. The lectin signal levels are indicated by color changes from yellow (high expression level) to blue (low expression level). (**B**) Comparison of the mean normalized signal of the 45 lectins between PC and NC. The NC and PC patients are indicated by blue and red, respectively. (**C**) Comparison of the mean normalized signal of the 45 lectins between pre- and post-operative sera. The pre- and post-operative sera are indicated by red and blue, respectively. Data are the means ± S.E. NC, normal controls; PC, pancreatic cancer; EV, extracellular vesicles.

**Figure 3 cancers-12-02469-f003:**
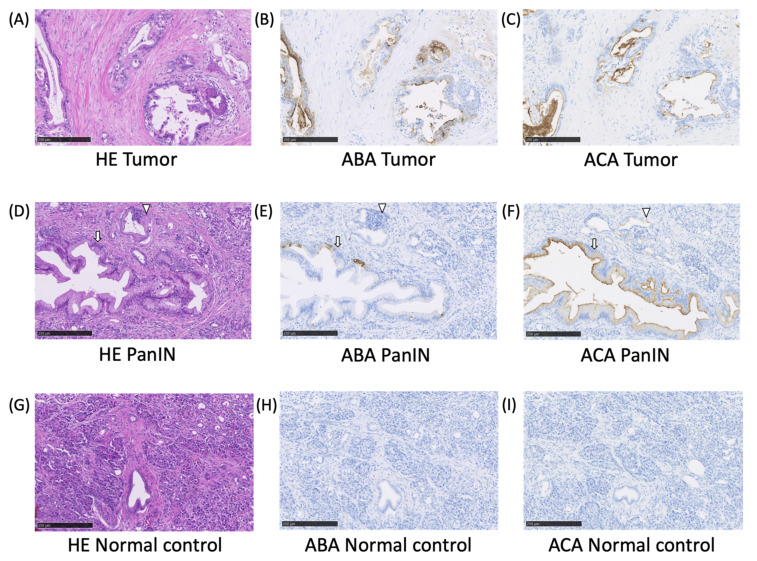
Histochemical staining of PC and PanIN lesions using ABA and ACA. Formalin-fixed paraffin-embedded tissue sections from a PC patient were stained with the biotinylated ABA and ACA. The upper panel shows the invasive tumor (**A**–**C**), and the lower panel shows PanIN (**D**–**F**), and the normal pancreatic tissue (**G**–**I**). Hematoxylin-eosin: A, D, and G, ABA: B, E, and H, ACA: C, F, and I. PanIN and acinar-to-ductal metaplasia are indicated with an arrow and an arrowhead, respectively. The scale bars show 250 µm.

**Figure 4 cancers-12-02469-f004:**
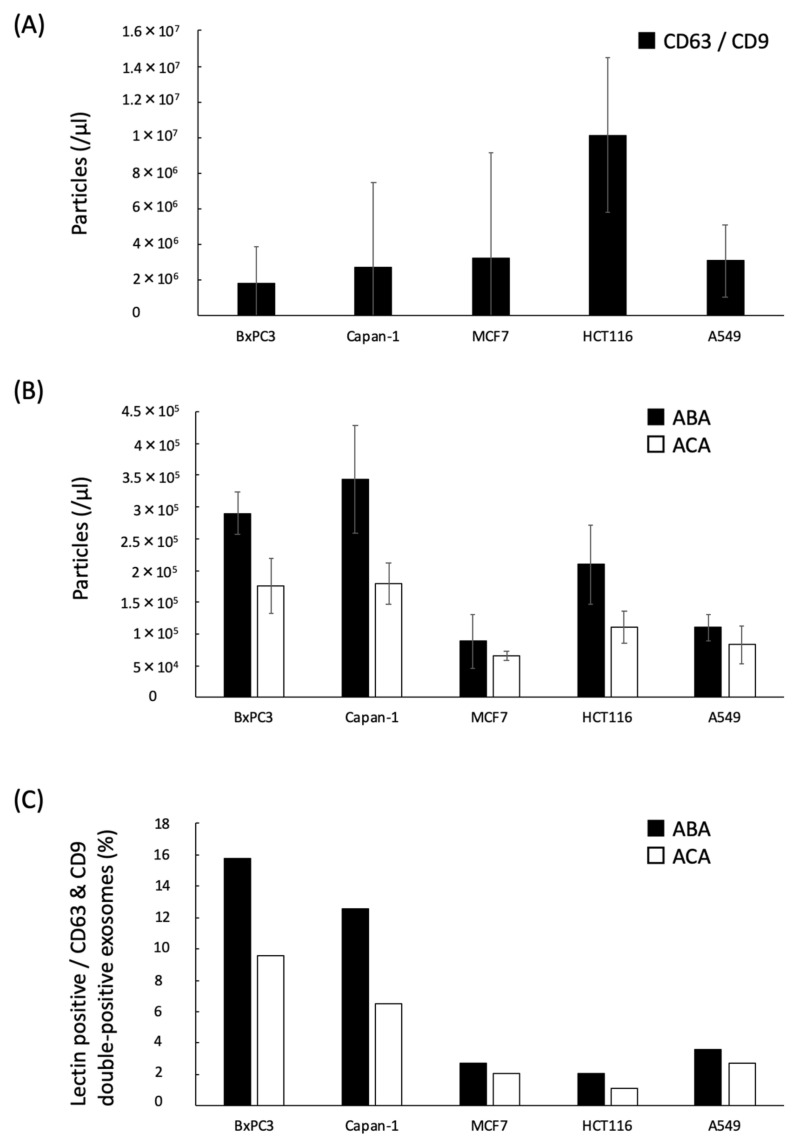
EVs derived from cancer cell cultures using ABA and ACA. EVs in the culture media with BxPC3, Capan-1, MCF7, HCT116, and A549 cells were quantified with ExoCounter. (**A**) Total EV detection using anti-CD63 Ab-coated disc and anti-CD9 Ab-conjugated beads. (**B**) Lectin-binding EV detection using the ABA- or ACA-coated disc and anti-CD9 Ab-conjugated beads. (**C**) Percentage of the ABA- or ACA-positive EVs normalized to CD63/CD9-positive EVs. Data are presented as means ± SD (*n* = 3).

**Figure 5 cancers-12-02469-f005:**
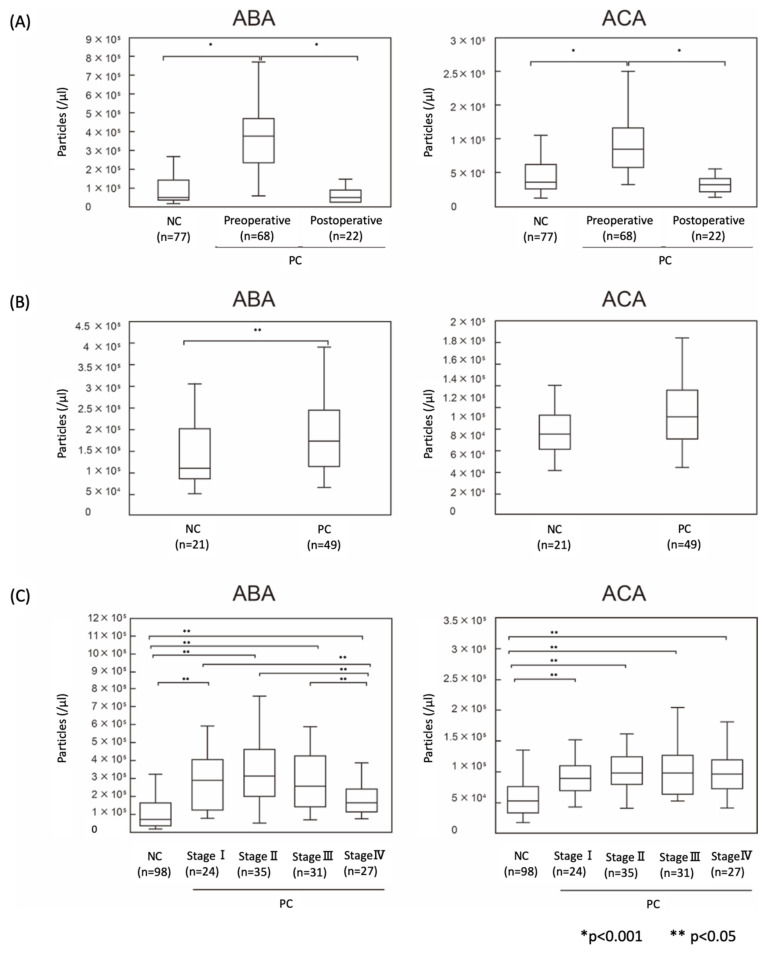
Quantification of ABA- and ACA-positive EVs in PC sera. Patient sera were analyzed using ABA- or ACA-coated discs and anti-CD9 Ab-conjugated beads with ExoCounter. (**A**) Quantification of ABA- and ACA-positive EVs from sera of preoperative and postoperative PC patients and NC (cohort 1). (**B**) Quantification using PC and NC sera obtained from cohort 2. (**C**) Comparison of ABA- and ACA-positive EVs from PC stage I–IV and NC sera in cohorts 1 and 2. Box plot indicates the 25th to 75th percentile. The medians of the samples are represented as the bars. p values were calculated using the Mann–Whitney U-test. * *p* < 0.05, ** *p* < 0.01.

**Figure 6 cancers-12-02469-f006:**
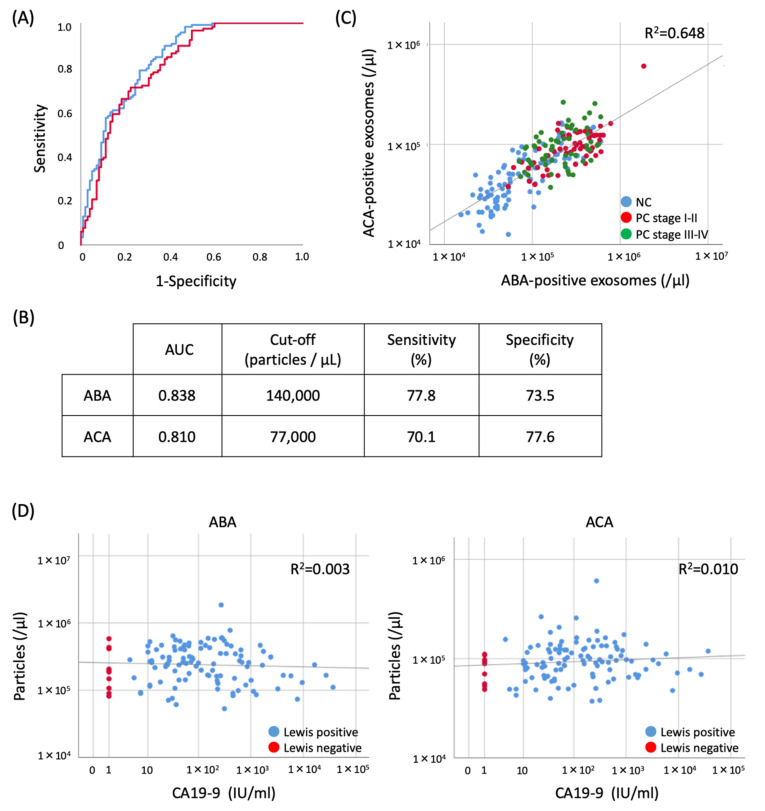
Characteristics of ABA- or ACA-positive EVs in PC sera. (**A**) The ROC curve obtained for ABA-positive EVs (blue line) and ACA-positive EVs (red line) differentiates PC patients from NC. (**B**) The AUC, a cut-off value, sensitivity, and specificity are described, respectively. (**C**) Scatter plots of ABA- and ACA-positive EVs in sera. Blue, red, and green dots indicate NC, PC stage I-II (early stage), and III-IV (advanced stage), respectively. (**D**) Scatter plots of the ABA- or ACA-positive EVs and CA19-9 in PC sera. Red and blue dots indicate Lewis antigen negative and positive patients, respectively.

**Table 1 cancers-12-02469-t001:** List of lectins with Altered Signal in PC Sera by lectin Microarray Analysis.

Lectins ^1^	Study 1				Study 2	
	Tumor/Normal Ratio	*p* Value ^2^			Pre-/Post-Operative Ratio	
DSA	1.375	<0.001	^3^	^4^	0.953	
STL	1.440	<0.001	^3^	^4^	0.822	
LEL	1.276	0.001	^3^	^4^	0.866	
UDA	1.170	0.011	^3^	^4^	0.780	
ACA	1.199	0.058	^3^	^4^	1.175	^6^
ABA	1.239	0.063	^3^	^4^	1.387	^6^
SSA	0.820	<0.001	^3^	^5^	1.097	
TxLC I	0.764	0.009	^3^	^5^	1.019	
Calsepa	0.800	0.012	^3^	^5^	1.054	
PHA(E)	0.814	0.085		^5^	0.959	
ACG	0.899	0.157				
LCA	0.898	0.186				
WGA	1.131	0.322				
AAL	1.119	0.443				

^1^ Lectins with a signal of Net Intensity >1000 (mean normalize > 0.2). ^2^ Student *t* test. ^3^ Significance by false discover rate using the Benjamini–Hochberg procedure. ^4^ Lectins significantly higher in cancer. ^5^ Lectins significantly lower in cancer. ^6^ Biomarker candidate.

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
