# Peer review of "O-Glycan-Altered Extracellular Vesicles: A Specific Serum Marker Elevated in Pancreatic Cancer"

_cancers, 2020, doi:10.3390/cancers12092469_

Round 1
Reviewer 1 Report
This is an interesting manuscript showing the first report of differential glycomic profiling of and quantification methods for cancer-specific exosomes derived from human PC sera. The present study utilized ExoCounter along with lectin microarray to identify potential biomarkers in a liquid biopsy for screening PC patients. The manuscript is well written, and experiments are well planned. Although I have a few concerns and believe to improve the manuscript further, the authors should take care of these:
The minor concerns are their figures which are unreadable more specifically Fig 2 and Fig 6 which are hard to read and importantly their whole manuscript is based on that.
Heatmap and bar-graph for study 2 are missing.
A little background on ABA and ACA and why they are elevated in PC would be beneficial to the reader.
What is the rationale not to include the 4 other Lectins in the discussion of the results which showed lower signal in PC for study 2.
Author Response
Responses to Reviewer #1.
This is an interesting manuscript showing the first report of differential glycomic profiling of and quantification methods for cancer-specific exosomes derived from human PC sera. The present study utilized ExoCounter along with lectin microarray to identify potential biomarkers in a liquid biopsy for screening PC patients. The manuscript is well written, and experiments are well planned. Although I have a few concerns and believe to improve the manuscript further, the authors should take care of these:
The authors would like to thank the reviewer for your constructive critique to improve the manuscript. We made every effort to address the issues raised and to respond to all comments. The revisions are indicated in red font in the revised manuscript. Please find next a detailed, point-by-point response to the reviewer's comments.
- The minor concerns are their figures which are unreadable more specifically Fig 2 and Fig 6 which are hard to read and importantly their whole manuscript is based on that.
We would like to thank the reviewer for the comment. Please note that we have used a readable font size in Figures 2 and 6 of the revised manuscript.
- Heatmap and bar-graph for study 2 are missing.
We would like to thank the reviewer for the comment. Please note that we have added the comparison bar-graph of the mean normalized signal of the 45 lectins between pre- and post-operative sera in Figure 2c.
- A little background on ABA and ACA and why they are elevated in PC would be beneficial to the reader.
We would like to thank the reviewer for the comment. We have added the following description of ABA and ACA in the Results section of the revised manuscript (marked with red).
“ABA and ACA are the O-glycan binding lectins (e.g., core 1 structure, Galα1-3GalNAc). Several studies have shown that hyper or altered O-glycosylation is known to be associated with carcinogenesis and tumor progression [16,17,22].” (Page 6, lines 182–185)
- What is the rationale not to include the 4 other Lectins in the discussion of the results which showed lower signal in PC for study 2.
We would like to thank the reviewer for the question. Besides ABA and ACA, four other lectins (DSA, STL, LEL, and DSA) were found to be candidates that recognize elevated exosomes in PC serum by lectin array analysis, but these elevations did not decrease in the postoperative serum. While further studies are needed to elucidate the exact reason for this, the altered glycosylations recognized by these lectins may be associated with inflammation in PC lesion. We have added these descriptions in the Discussion section of the revised manuscript as follows:
“As the blood contains EVs derived from the tumor lesion and healthy tissues, the background value is relatively high, and therefore, precise differential analysis is required. As the lectin array that screened the specific glycans of EVs was performed by qualitatively analysis, the relative signal difference level was low. By using the ExoCounter that can detect absolute quantification of specific EVs, we could elucidate a significant difference in PC sera for the candidate lectin (ABA and ACA)-positive EVs.”
(Page 11, lines 306-311).
Reviewer 2 Report
In the manuscript titled O-Glycan-Altered Exosomes: A Specific Serum Marker Elevated in Pancreatic Cancer, Yokose et al. profiled the glycome of exosomes in the serum sample from pancreatic cancer patients and identified two glyco-candidates as biomarkers. The identification of biomarker for pancreatic cancer is of high significance as it will facilitate the early diagnosis of the disease, which is critical for better prognosis. Although the ABA and ACA looks promising biomarkers, from the data presented they were only marginally higher in pancreatic cancer than that in normal control. As a result, the sensitivity and specificity is still less than optimal. Nonetheless, the application of ABA and ACA could provide additional prediction power for pancreatic cancer.
Some comments:
- Figure 2A, the clustering of lectins can be improved. For example, ConA and ACA is obviously different than other neighbors. Different clustering methods should be implemented to improve the clustering.
- Figure 3, a normal control is needed to demonstrate the specificity of the antibodies.
- Figure 4, how many replicates of this experiment were performed? There was no error bar for all columns. Was it because the experiment was done once?
Author Response
Responses to Reviewer #2
In the manuscript titled O-Glycan-Altered Exosomes: A Specific Serum Marker Elevated in Pancreatic Cancer, Yokose et al. profiled the glycome of exosomes in the serum sample from pancreatic cancer patients and identified two glyco-candidates as biomarkers. The identification of biomarker for pancreatic cancer is of high significance as it will facilitate the early diagnosis of the disease, which is critical for better prognosis. Although the ABA and ACA looks promising biomarkers, from the data presented they were only marginally higher in pancreatic cancer than that in normal control. As a result, the sensitivity and specificity is still less than optimal. Nonetheless, the application of ABA and ACA could provide additional prediction power for pancreatic cancer.
The authors would like to thank the reviewer for your constructive critique to improve the manuscript. We made every effort to address the issues raised and to respond to all comments. The revisions are indicated in red font in the revised manuscript. Please find next a detailed, point-by-point response to the reviewer's comments.
Some comments:
- Figure 2A, the clustering of lectins can be improved. For example, ConA and ACA is obviously different than other neighbors. Different clustering methods should be implemented to improve the clustering.
We would like to thank the reviewer for the comment. By using the normalized data, we re-evaluated the HCA of lectin array results. Thereby, the lectin clustering improved. We have modified the clustering data of Figure 2a in the revised version.
- Figure 3, a normal control is needed to demonstrate the specificity of the antibodies.
We would like to thank the reviewer for the comment. In the histochemical analyses, we used the biotinylated lectins. To show the specificity of this staining, we have added the lectin-staining images of the normal area in PC tissues in Figures 3g, h, and i of the revised manuscript.
- Figure 4, how many replicates of this experiment were performed? There was no error bar for all columns. Was it because the experiment was done once?
We would like to thank the reviewer for the comment. This experiment was performed in triplicate. In the revised version, we have modified the graphs of the mean ± SD from three independent experiments in Figures 4a and b.
Reviewer 3 Report
In this manuscript, Kitakawa and colleagues have performed for the first time a glycomic profile of extracellular vesicles derived from the serum of pancreatic cancer (PC) patients using a lectin-based microarray. The authors identified the O-glycan-binding lectins ABA and ACA as potential candidates for liquid biopsy in PC patients. Using Exocounter, a novel device for absolute quantification of exosomes, the authors found that the quantification of serum-derived exosomes recognized by ABA and ACA lectins provided a good discrimination between PC patients and healthy controls proposing these lectins as candidates for liquid biopsy in PC diagnosis. Remarkably, surgical resection of tumors decreased the levels of studied glycans in serum-derived exosomes showing a good correlation of these biomarkers with tumor burden levels.
Despite the novelty of the determination of glycomic profiles in exosomes in clinical samples and the fact that the manuscript is be one of the first works using a novel device for exosome quantification, several concerns raise over these strong points of the work:
- The lectin microarray did not provide a proper discrimination among PC patients and controls. As shown in Figure 2, hierarchical clustering did not stratify correctly all the samples and the same conclusion can be observed in the Principal Component Analysis (PCA) shown in the supplementary figure 1, a proper description of this point in the text is missing. To reinforce the informative character of this microarray, it is advisable to filter the lectins by expression levels or select the lectins subgroup that could allow a clear separation among the control and patient groups. Beside that, the signal ratios for tumor versus normal for the most differentially detected lectins (1.199 for ACA and 1.239 for ABA) are low and not reassuring at all, particularly when taking into account the intra and inter-group variability occurring in clinical samples. It would be desirable to indicate the intra-group coefficient of variation of the study.
- The techniques used for exosome purification (Magcapture and Exocounter) are relatively new and not validated by many other groups. In fact, the references to Exocounter come from the same group that present this manuscript. These facts demand a more detailed description of the methods in the main text in addition to the technical description already present in the methods section. Additionally, a validation of some of the findings by other exosome purification method would add reliability to the data.
- Related to the previous point, it is unclear how authors discriminated microvesicles from exosomes. CD9 and CD63 used in the Exocounter are also expressed in other vesicle types. Similarly phospholipids recognized by TIM-4 are present in other vesicle types. If a previous size separation procedure is not performed, how the authors are convinced that they are only measuring exosomes? NTA tracking system is biased towards sizes around 100nm even if larger vesicles are injected in the device. Transmission electron microscopy validation is required. I suggest a verification of some of their findings by other more standard methods for exosome isolation and the use of additional exosome markers apart from CD9 and CD63.
- The authors detected differences in ACA and ABA lectins in early stage PC patients including CA19-9 negative cases suggesting the advantage of using these biomarkers for early detection of PC including the detection of false negatives currently misdiagnosed because the absence of CA19-9 in the Lewis negative phenotype of PC. However, the authors did not address the problem of false positives caused by other diseases such as chronic pancreatitis, bile duct stones or other gastrointestinal malignancies that display increased CA19-9 levels. Measurement of ACA and ABA positive extracellular vesicles in some of these patients would shed light about the potential use of the proposed biomarkers in early PC diagnosis.
Additional comments:
- Authors affirmed in the introduction that “not reliable PC-specific exosome marker has not yet been identified”. However, several miRNAs and Glypican-1 have been described as potential biomarkers for PC patients. Those references should be included.
- In table 1, significance should be expressed in FDR terms as it refers to a multiple comparison analysis.
- In figure 3, Images need more magnification and proper healthy tissues should be presented. In the tiny areas of adjacent healthy tissue shown in the pictures, it is not possible to appreciate if normal ducts also are reactive to ACA and ABA.
- Figure 5c, p-values are missing in most of the comparisons.
- Supplementary Figure 1: Plots are lacking any quantitative information on the vertical and horizontal axes.
- Some additional editing in English is required.
Author Response
Responses to Reviewer # 3
In this manuscript, Kitakawa and colleagues have performed for the first time a glycomic profile of extracellular vesicles derived from the serum of pancreatic cancer (PC) patients using a lectin-based microarray. The authors identified the O-glycan-binding lectins ABA and ACA as potential candidates for liquid biopsy in PC patients. Using Exocounter, a novel device for absolute quantification of exosomes, the authors found that the quantification of serum-derived exosomes recognized by ABA and ACA lectins provided a good discrimination between PC patients and healthy controls proposing these lectins as candidates for liquid biopsy in PC diagnosis. Remarkably, surgical resection of tumors decreased the levels of studied glycans in serum-derived exosomes showing a good correlation of these biomarkers with tumor burden levels.
Despite the novelty of the determination of glycomic profiles in exosomes in clinical samples and the fact that the manuscript is be one of the first works using a novel device for exosome quantification, several concerns raise over these strong points of the work:
The authors would like to thank the reviewer for his/her constructive critique to improve the manuscript. We made every effort to address the issues raised and to respond to all comments. The revisions are indicated in red font in the revised manuscript. Please find next a detailed, point-by-point response to the reviewer's comments.
- The lectin microarray did not provide a proper discrimination among PC patients and controls. As shown in Figure 2, hierarchical clustering did not stratify correctly all the samples and the same conclusion can be observed in the Principal Component Analysis (PCA) shown in the supplementary figure 1, a proper description of this point in the text is missing. To reinforce the informative character of this microarray, it is advisable to filter the lectins by expression levels or select the lectins subgroup that could allow a clear separation among the control and patient groups. Beside that, the signal ratios for tumor versus normal for the most differentially detected lectins (1.199 for ACA and 1.239 for ABA) are low and not reassuring at all, particularly when taking into account the intra and inter-group variability occurring in clinical samples. It would be desirable to indicate the intra-group coefficient of variation of the study.
We would like to thank the reviewer for the comment. Regarding lectin array analyses, NC and PC could not be further stratified by re-filtering or re-clustering. As the blood contains EVs derived from the tumor lesion and healthy tissues, the background value is relatively high, and therefore, precise differential analysis is required. As the lectin array that screened the specific glycans of EVs was performed by qualitatively analysis, the relative signal difference level was low. By using the ExoCounter that can detect absolute quantification of specific EVs, we could elucidate a significant difference in PC sera for the candidate lectin (ABA and ACA)-positive EVs. These descriptions have been added in the Discussion section of the revised manuscript (Page 11, lines 306–311).
- The techniques used for exosome purification (Magcapture and Exocounter) are relatively new and not validated by many other groups. In fact, the references to Exocounter come from the same group that present this manuscript. These facts demand a more detailed description of the methods in the main text in addition to the technical description already present in the methods section. Additionally, a validation of some of the findings by other exosome purification method would add reliability to the data.
We would like to thank the reviewer for the comment. We have added the detail descriptions of MagCapture and ExoCounter analysis, as per the reviewer’s suggestions.
“In brief, 0.6 mg of streptavidin magnetic beads, bound with 1 µg of biotinylated mouse Tim4-Fc, was added to serum supplement and rotated overnight at 4°C. The beads were washed three times with 1 ml of washing buffer (20 mM Tris-HCl, pH 7.4, 150 mM NaCl, 0.0005% Tween20, 2 mM CaCl2), and the bound EVs were eluted with elution buffer (20 mM Tris-HCl, pH 7.4, 150 mM NaCl, 2 mM EDTA) [43].” (Page 13, lines 389–393)
“Quantification of EVs using ExoCounter presents several advantages (e.g., the ability to quantify the absolute number of EVs containing surface antigen high sensitivity and high linearity) compared to conventional methods, such as ELISA or flowcytometry [33,34].” (Page 11, lines 287–289)
- Related to the previous point, it is unclear how authors discriminated microvesicles from exosomes. CD9 and CD63 used in the Exocounter are also expressed in other vesicle types. Similarly phospholipids recognized by TIM-4 are present in other vesicle types. If a previous size separation procedure is not performed, how the authors are convinced that they are only measuring exosomes? NTA tracking system is biased towards sizes around 100nm even if larger vesicles are injected in the device. Transmission electron microscopy validation is required. I suggest a verification of some of their findings by other more standard methods for exosome isolation and the use of additional exosome markers apart from CD9 and CD63.
We would like to thank the reviewer for the comment. As you pointed out, analyses with the ExoCounter system can detect exosomes excluding microvesicles because the groove on the disc that captures particles is 100 nm in size, but analyses with lectin array using particles isolated with MagCapture could contain microvesicles. Therefore, we have changed the term from “exosomes” to “extracellular vesicles (EVs)” in the revised manuscript. In addition, we have confirmed the purity of particles isolated with MagCapture using STEM. We have added the NTA data and STEM images of the isolated vesicles in Figure S1a and b.
- The authors detected differences in ACA and ABA lectins in early stage PC patients including CA19-9 negative cases suggesting the advantage of using these biomarkers for early detection of PC including the detection of false negatives currently misdiagnosed because the absence of CA19-9 in the Lewis negative phenotype of PC. However, the authors did not address the problem of false positives caused by other diseases such as chronic pancreatitis, bile duct stones or other gastrointestinal malignancies that display increased CA19-9 levels. Measurement of ACA and ABA positive extracellular vesicles in some of these patients would shed light about the potential use of the proposed biomarkers in early PC diagnosis.
We would like to thank the reviewer for the comment. As you pointed out, we have not determined whether the ABA or ACA reacts with the false-positive samples, such as chronic pancreatitis, bile duct stones, or other gastrointestinal malignancies. These analyses would further support to demonstrate the utility of our system in the future. Please note that we have added the following description in the Discussion section of the revised manuscript.
“We have not examined the false-positive cases, in which CA19-9 is elevated in other diseases. Further investigation is needed using CA19-9 false positives cases with other diseases, such as chronic pancreatitis, bile duct stones, or other gastrointestinal malignancies, that display increased CA19-9 levels.” (Page 10, line 329–page 11, line 337).
Additional comments:
Authors affirmed in the introduction that “not reliable PC-specific exosome marker has not yet been identified”. However, several miRNAs and Glypican-1. have been described as potential biomarkers for PC patients. Those references should be included.
We would like to thank the reviewer for the additional comment. Please note that we have added further information obtained from previous studies including miRNAs and glypican-1 in the Introduction section of the revised manuscript.
“In PC patients, several miRNAs in EVs have been used as useful biomarkers [30]. Recently, the Glypican-1-positive EVs have been reported to be elevated in early PC serum [31]; however, other group analyses did not show any changes in glypican exosome in pancreatic cancer [32]. Furthermore, a PC-specific glycomic alteration on EVs has not been identified.” (Page 3, lines 97–99)
- In table 1, significance should be expressed in FDR terms as it refers to a multiple comparison analysis.
We would like to thank the reviewer for the comment. We have verified the lectin array data by the Benjamini-Hochberg method using FDR, and we have showed the lectins with a significant level in Table 1. We have added the statistical results and footnote in Table 1 of the revised manuscript.
- In figure 3, Images need more magnification and proper healthy tissues should be presented. In the tiny areas of adjacent healthy tissue shown in the pictures, it is not possible to appreciate if normal ducts also are reactive to ACA and ABA.
We would like to thank the reviewer for the comment. We have modified more broad area of tissue images, and also added the lectin-staining images of normal area in PC tissues in Figures 3g, h, and i of the revised version.
- Figure 5c, p-values are missing in most of the comparisons.
We would like to thank the reviewer for the comment. Please note that we have added the p-values in Figure 5c.
- Supplementary Figure 1: Plots are lacking any quantitative information on the vertical and horizontal axes.
We would like to thank the reviewer for the comment. Please note that we have plotted the intrinsic values on the vertical and horizontal axes.
- Some additional editing in English is required.
We would like to thank the reviewer for the comment. We have provided our manuscript to a proofreading company for additional English editing (Editage).
Reviewer 4 Report
Majors:
- Histochemical staining using biotinylated ABA or ACA should be performed also in normal controls
- Size distribution could be misleading. Isolated extracellular vesicles should be further confirmed by dynamic light scattering
- Further investigation of nucleic acids information (DNA, RNA, miRNA) is strongly encouraged
- Within the discussion, limitations and clinical implications (such as difficult shedding of pancreatic cancers) must be more extensively pointed out.
Minors:
- The term “exosomes” must be reconsidered, since tracing such vesicles back to a particular biogenesis pathway is difficult. The term “Extracellular vesicles” is strongly suggested.
- Do update epidemiology references (e.g.: Siegel et al 2020)
- Clinical-pathological characteristics among cohorts should be more precisely balanced
Author Response
Responses to Reviewer # 4
The authors would like to thank the reviewer for your constructive critique to improve the manuscript. We made every effort to address the issues raised and to respond to all comments. The revisions are indicated in red font in the revised manuscript. Please find next a detailed, point-by-point response to the reviewer's comments.
Major:
- Histochemical staining using biotinylated ABA or ACA should be performed also in normal controls
We would like to thank the reviewer for the comment. We have added the lectin-staining images of normal area in PC tissues in Figures 3g, h, and i of the revised version.
- Size distribution could be misleading. Isolated extracellular vesicles should be further confirmed by dynamic light scattering
We would like to thank the reviewer for the comment. We have confirmed the purity of particles isolated with MagCapture using STEM. We have added the NTA data and STEM images of the isolated vesicles in Figure S1a and b.
- Further investigation of nucleic acids information (DNA, RNA, miRNA) is strongly encouraged
We would like to thank the reviewer for the comment. The correlation analysis between the o-glycans alteration and nucleic acid information (DNA, RNA, miRNA) is an interesting subject, and we would like to analyze it in the future. We have made corrections in the revised manuscript as follows:
“The correlation analysis between the O-glycan alterations and nucleic acid information including DNAs or miRNAs is an interesting subject, and further investigation is needed in the future.” (Page 12, lines 326–328)
- Within the discussion, limitations and clinical implications (such as difficult shedding of pancreatic cancers) must be more extensively pointed out.
We would like to thank the reviewer for the comment. We have added the description of limitations and clinical implications of our study in the Discussion section of the revised manuscript as follows:
“However, there were some limitations in the current study. 1) All of the samples examined are of PC with a diagnosis, and cases of undiagnosed early PC have not been verified. Further verification is required for these cases. 2) We have not examined the false-positive cases, in which CA19-9 is elevated in other diseases. Further investigation is needed using CA19-9 false positives cases with other diseases, such as chronic pancreatitis, bile duct stones, or other gastrointestinal malignancies, that display increased CA19-9 levels. 3) Although we used two independent cohorts, our participants were only Japanese, which may have resulted in potential selection bias. In addition, there was no clinicopathologic background matching between the two cohorts. 4) Serum specimens in this study were not collected from age-matched controls and PC patients.” (Page 11, line 329–page 12, line 337)
Minors:
- The term “exosomes” must be reconsidered, since tracing such vesicles back to a particular biogenesis pathway is difficult. The term “Extracellular vesicles” is strongly suggested.
We would like to thank the reviewer for the comment. We have revised the term from “exosomes” to “extracellular vesicles,” as we could not distinguish between them appropriately.
- Do update epidemiology references (e.g.: Siegel et al 2020)
We would like to thank the reviewer for the comment. Following the reviewer’s suggestion, we have modified the update epidemiology reference in the revised manuscript.
- Clinical-pathological characteristics among cohorts should be more precisely balanced
We would like to thank the reviewer for the comment. We have made corrections in red and underlined them in the main manuscript. The revisions are as follows:
“3) Although we used two independent cohorts, our participants were only Japanese, which may have resulted in potential selection bias. In addition, there was no clinicopathologic background matching between the two cohorts. 4) Serum specimens in this study were not collected from age-matched controls and PC patients
.” (Page 12, lines 334–337)
Round 2
Reviewer 3 Report
The authors have substantially improved the background and clarity of the scientific message of the manuscript. They also have answered all the risen concerns. I congratulate the authors and consider the manuscript suitable for publication in this journal.
Reviewer 4 Report
Major flaws have been properly implemented.